# Rabies Vaccination for Sheep and Goats: Influence of Booster on Persistence of Antibody Response

**DOI:** 10.3390/vetsci11100502

**Published:** 2024-10-13

**Authors:** Sarah Weyl Feinstein, Shiri Novak, Marina Eyngor, Yaniv Lavon, Boris Yakobson

**Affiliations:** 1Western Galilee Veterinary District, Veterinary Services, Ministry of Agriculture and Food Security, Bet Dagan, Rishon Letzion 7519701, Israel; 2Rabies Department, Kimron Veterinary Institute, Bet Dagan 5025001, Israel; shirin@moag.gov.il (S.N.); marinal@moag.gov.il (M.E.); borisy@moag.gov.il (B.Y.); 3Israel Cattle Breeders’ Association, Caesarea Industrial Park, Caesarea 38900, Israel; yaniv@icba.co.il

**Keywords:** rabies vaccination, ewes, does, antibodies, booster

## Abstract

**Simple Summary:**

Rabies is causing a serious threat to humans and animals almost worldwide. Immunization against the disease is important in four aspects: public health protection, animal health, economic impact, and regulatory compliance. Rabies affects a wide range of mammals, including domestic pets, livestock, and wild animals. Infected animals, usually bitten by an infected wild animal, suffer from severe neurological symptoms and eventually die. In Israel, mandatory vaccination of dogs as well as wildlife surveillance exists. When it comes to farm animals, the vaccination policy depends on a risk analysis. However, little is known regarding the optimal immunization protocol needed to protect small ruminants. To find an optimal vaccination strategy, we tested the effect of one and two doses of vaccine on sheep and goats. Initially, animals were vaccinated with one dose, and half of them received a second dose (booster) a month later. The level of antibodies was tested a month and a year later. The study demonstrated a different immune response between the different species, as one-dose vaccinated sheep showed a low maintenance of protection a year later, compared with the booster-receiving group. On the other hand, 80% of the goats which received a single dose had a sufficient antibody level a year later.

**Abstract:**

Infrequent rabies cases occur in Israel, endangering humans and animals. While dogs receive mandatory vaccinations, farm animals are vaccinated voluntarily. However, optimal vaccination protocol for small ruminants is lacking. The aim of this study was to test the immunological responses to the rabies vaccine, with or without a booster, in sheep and goats; 70 ewes and 49 does participated in the trial. Following the first vaccine, 88% of the ewes and 100% of the does had a sufficient level of rabies antibodies (>0.5 IU/mL) 30 days post-vaccination. A year later, 82% of the ewes that had received a booster dose remained protected, whereas 46% of the non-boosted ewes had a sufficient antibody level. For does, 83% of those receiving a booster maintained sufficient antibody levels 1 year later; 80% of the non-boosted does remained protected, demonstrating no significant contribution of the booster dose in this group of goats. However, while the initial immunological response of the does was higher, the change in response between 1 month and 12 months post-vaccination differed significantly between species, with a greater titer reduction in the does. Differential immunological responses between individuals and between species warrant longer-term studies to recommend a proper vaccine protocol for each species.

## 1. Introduction

Rabies is an acute, fatal, neurological disease that affects all different types of mammals. This important disease is caused by the rabies virus, a negative-sense single-stranded RNA virus in the genus *Lyssavirus* of the family *Rhabdoviridae* [1]. Today, rabies continues to pose a serious public health threat in most areas of the world, especially in developing countries in Asia and Africa. The World Health Organization (WHO) estimates that, annually, more than 59,000 human deaths are caused by rabies worldwide due to rabid animal bites [2].

The Middle East region—including Saudi Arabia, Oman, Yemen, Iran, Turkey, and Israel—is facing occurrences of sylvatic rabies, usually involving the golden jackal (*Canis aureus*) and the red fox (*Vulpes vulpes*) [3]. In Israel, rabies is endemic and exists mainly in wild jackals, but the virus also infects stray dogs, foxes, wolves, and livestock. Wildlife rabies in Israel has been controlled at a national level since 1998 by using oral rabies vaccines, which are distributed in the environment [4]. Infrequent cases in golden jackals and stray dogs occur as spillover from infected animals or at the borders with other countries that are endemic for rabies. Vaccination, microchip identification, and registration are mandatory for dogs at a national level, with records kept in an electronic database. During outbreaks, most cases arise from carnivores, but livestock can be infected as well, especially in the north of Israel where they live in pastures [5]. For example, a large outbreak occurred from January 2017 to April 2018, with four times more cases than the annual average. In this outbreak, 85 positive cases were found in the northeastern part of Israel, close to the border with Jordan. Among the sick animals were 20 cases (23%) of infected farm animals [6]. In addition to the public health risk, there are economic consequences due to the death of livestock and quarantine restrictions.

Compared to rabies surveillance in wild animals, rabies control in farm animals is much easier to implement. Prophylactic vaccination against rabies is recommended for farm animals in endemic regions [7]. In Israel, rabies vaccination is voluntary for livestock, and mandatory for those that are publicly exhibited. Vaccination schedules for farm animals vary based on farm practices and perception of risk. Following the 2017–2018 outbreak, Israel Veterinary Services established a mandatory livestock vaccination policy in risk areas, due to the potential exposure of livestock to infected wild jackals. Many of the farms do not have fences or closed gates, and animals are held in pasture, where the potential exposure increases. Moreover, the exposed farm animals can further endanger the caretakers—a dangerous risk.

Regarding small ruminants, no recommendations for using a booster when vaccinating small ruminants was found. However, Kimron Veterinary Institute and the Koret School of Veterinary Medicine faculty have investigated the persistence of anti-rabies antibodies in both cattle and horses; the first study testing the rabies vaccination protocol in cattle revealed that most cattle older than 6 months are protected after a single inoculation, and a second inoculation ensures higher antibody levels for improved protection. Nevertheless, cattle receiving an effective priming dose responded well to a booster up to 36 months later [8]. The results of a study on horses demonstrated that over 90% of horses retain their protective immunity against rabies for more than 1 year and 70% of them for up to 8 years following vaccination [9]. Practical applications arising from those studies rely on the assumption that routine vaccination every 2 or 3 years instead of annual vaccination could reduce labor and expenses.

The aim of the current study was to examine the immunological response of sheep and goats to the different rabies vaccination protocols, to better understand the immunological response to the vaccine and recommend a more adequate vaccination protocol.

## 2. Materials and Methods

### 2.1. Study Design

Two private commercial herds in northern Israel participated in the study:A sheep fattening farm, located in moshav Kfar Hasidim, near Haifa, with 600 mixed-breed adult ewes.A mixed-alpine goat dairy farm, located in the Manof school in Acre, with 230 adult does.

The study was performed under veterinary supervision with animal care and use approval from the Ethics Committee of the Volcani Institute, Agricultural Research Organization, Ministry of Agriculture.

All animals were identified by official tags, in addition to their internal farm tags. The animals participating in the study were born on the farms and were not vaccinated against rabies before the experiment. Preventive health measures for the herds included FMD vaccination, and all other treatment records were maintained for the life of each animal. For this study, a group of 70 pregnant ewes (1 year of age) were randomly selected. Following the sale of animals for meat production, in the following year, only 30 sheep from that group continued to participate in the study. From the goat farm, 49 pregnant does participated in the first year, among which 27 were further tested in the following year.

All participants were serologically tested for rabies antibodies at the start of the study before they received the first dose of rabies vaccine. A month (4 weeks) after the first dose, blood samples were taken to determine the levels of rabies antibodies. After this sampling, half of each group received a booster injection, and the other half did not. The change in antibody titer was evaluated 1 year later in all individuals. In addition, 20 female goat kids from vaccinated mothers were tested for their antibody levels at a mean age of 58 ± 7 days.

### 2.2. Vaccination and Sampling

The rabies vaccine was administered using the commercial “Rabisin”, a licensed killed-virus and adjuvanted rabies vaccine (Boehringer Ingelheim), at the recommended 1.0 mL dose, delivered intramuscularly [10]. Blood was collected from the jugular vein into glass clotting tubes. Serum was separated by centrifugation following incubation at room temperature for 3 h. Sera to be tested in the same week of collection were stored at 4 °C. Sera to be tested later were stored at −20 °C until processing. Additionally, we tested the antibody level in the serum of 20 kids born to the vaccinated does at the age of 2 months, and nine colostrum samples of vaccinated ewes were examined for antibody concentration. These kids received colostrum from a “pull-colostrum bank” containing colostrum from vaccinated and non-vaccinated ewes.

### 2.3. Laboratory Methods

Samples were tested for rabies antibodies at Kimron Veterinary Institute using the Rabies Fluorescent Focus Inhibition Test (RFFIT) [11]. Sera were analyzed for the presence of rabies virus-neutralizing antibodies using the RFFIT. Animals with antibody titers ≥ 0.5 IU/mL were considered to be protected against rabies. Titer levels were recorded as serial dilutions with positive and negative control samples added at each session. The RFFIT is considered the gold standard by the WHO [12] and is shown to have a sensitivity and specificity of 100% and 89–100%, respectively [13]. Serological titers were converted to international units per milliliter, resulting in a linear scale of measurement in which a titer value of >0.5 IU/mL was considered a protective antibody level.

### 2.4. Statistical Analysis

Results are presented as mean ± standard error (SE). Differences in protection rates by different factors were tested using the Fisher exact test (two-tailed). The factors associated with antibody level after 1 month and 1 year were analyzed in multifactorial models using the proc glimmix procedure of SAS (version 9.2, SAS Institute, Cary, NC, USA). The statistical model included antibody level after 1 month or after 1 year (success) = species + booster + species*booster, where species = ewes (sheep) or does (goats), booster = received or did not receive, and success = level of antibody higher or lower than 0.5 IU/mL. Due to the low number of animals remaining for the analysis after 1 year, we ran a post hoc power analysis.

To compare levels within a variable, we ran the Bonferroni adjustment for multiple comparisons. The figures present the average of the data per species, the tables present the least squares mean (LSM) difference between the parameters ± standard error of the mean (SEM), and *p* values are presented for each comparison.

## 3. Results

### 3.1. Ewes’ Immunological Response—Comparing 1 Month to 12 Months Post-Vaccination and Boosted vs. Non-Boosted

The overall effect of the vaccine, with and without a booster, can be seen in Table 1 and Figure 1. Following the first dose for all ewes, the mean antibody level was 2.79 ± 2.4, and the median was 1.45 IU/mL. Values for these 70 ewes ranged from 0.2–33 IU/mL. However, eight individuals had an insufficient antibody level, i.e., less than 0.5 IU/mL. Thus, 88% of the ewes were in the protected group. On the other hand, two ewes had 14 IU/mL and one had 33 IU/mL of rabies antibodies. A year later, regardless of booster status, 20 out of 30 ewes had a serological protective level of antibodies, meaning overall only 66% were protected (Fisher extract: *p* = 0.04). Surprisingly, among the seventeen re-vaccinated ewes, three did not have a sufficiently protective level of antibodies, meaning only 82% of the booster-receiving group were immunologically protected. Two of the unprotected ewes had a good level the previous year (one had 14 IU/mL), and the third had an insufficient level the year before as well. Nevertheless, the average level following the booster was 6.95 ± 12 IU/mL and the median was 1.4 IU/mL. The mean result of the group that was not re-vaccinated a month later was 1.95 ± 3.5 IU/mL, with a median of 0.5 IU/mL. Of the thirteen ewes that did not receive the booster dose, only six (46%) had enough protection, while the other seven animals (54%) had insufficient antibody level (Table 1).

### 3.2. Goats’ Immunological Response—Comparing 1 Month to 12 Months Post-Vaccination and Boosted vs. Non-Boosted

The results 1 month and 1 year after the rabies vaccination are provided in Figure 2 and Table 2. The does’ initial immunological response in this study was more prominent than the ewes’ response (Figure 2). All does had values above 0.5 IU/mL, so they were all sufficiently protected after 1 month. The mean antibody level of the 49 does following the first dose was 12 ± 9 IU/mL. The range of response after 1 month was 1.3–46.4 IU/mL, and the median was 8 IU/mL. They did not all maintain a sufficient antibody level 1 year later. The group that did not receive a booster maintained a mean level of 2.38 ± 3.6 IU/mL, whereas the booster group’s level was 3.67 ± 3.4 IU/mL. Regardless of booster, five of the twenty-seven goats had an insufficient antibody level after 1 year. Out of twelve does who received the booster dose, only two failed to have a minimally protective antibody level. However, out of the 15 animals that did not receive the booster, 12 maintained a good antibody level. Overall, 83% of the does receiving the booster maintained a sufficiently protective antibody level 1 year later, compared to 80% of the goats without the booster (not significant).

### 3.3. Comparison between Species

Comparing sheep’s and goats’ serological results 1 month post-vaccination, there was a significantly higher mean antibody titer in does compared to ewes (Figure 3 and Table 3). The difference between species was not significantly different 1 year later (Table 4). However, the difference between sampling points (1 month vs. 1 year) for each species was significant, due to the different immunological responses in these two species. Success in maintaining immunity was also significantly different (*p* = 0.04), with the goats having a high response to the first vaccine that decreased 1 year later, whereas in sheep, the initial response was lower and increased slightly 1 year later (Figure 3). After 1 year, the mean difference in ewe and goat antibody levels was 2.13 and −9.8, respectively.

### 3.4. Immunological Level of Antibodies in Female Kids of Vaccinated Goats

We tested the antibody level in the serum of kids born to the vaccinated does at the age of 2 months. Their antibody levels ranged from 0.03–0.78 IU/mL. Of these 20 female kids, 5 had a sufficient level of rabies antibodies (defined as values > 0.5 IU/mL).

### 3.5. Level of Antibodies in the Colostrum of Vaccinated Does

We tested the rabies antibody level in the colostrum of nine vaccinated does 8.7 ± 6 days after parturition. The does were all pregnant when they received the first vaccine. The antibody level of all samples was below the protective level, with a mean of 0.16 ± 0.1 IU/mL. Although adequate levels of antibodies were not found in the colostrum, two kids of the tested mothers had protective titers.

## 4. Discussion

The serological immune response post rabies vaccination in sheep and goats was generally strong when measured in terms of virus-neutralizing antibody levels. Nevertheless, the immunological response differed significantly between the two species. In does, it was much more prominent after 1 month, then decreased 1 year later. In contrast, the ewes had lower antibody levels at 1 month, but these remained approximately the same 1 year later. Regardless of the booster effect, 66% of ewes and 82% of does were protected 1 year after the first vaccination. Testing persistency a year later, 82% of the ewes and 83% of the does that had received a booster dose maintained sufficient antibody level. We are aware of the unsignificant result in the immune response after 1 year in contrast to the significant difference we found after 1 month. The reason for that could be due to the low number of animals that remained on the farms because of management reasons. Because of the low number of animals, we ran a post hoc power analysis for the data after 1 year and which showed a power of 0.843, which is not so strong, but still falls within an acceptable range. Taking into account the limitation of this study because of the low number of animals, we still think that the result is important and will help us to achieve higher protection rates in the future.

The novelty of this study arises from the examination of the antibody persistence following rabies vaccine in small ruminants, which was not tested until now. For comparison, when examining the immunological response of dairy cattle following a booster after 2 years from vaccination, 90% of cows had sufficient antibody levels [14], and in the Yakobson et al. study, 2–3 years post-vaccination, 60% of the cows remained immune [8]. The Blancou et al. study, which studied the persistence of antibodies following bovine vaccination for 55 months, demonstrated that the sufficient titers remained and increased 16 months after the booster injection from the primary immunization [15].

Several factors are known to influence the strength and duration of the rabies immune response in ruminants: the possible presence of maternally derived antibodies, the age of the inoculated animal, and the number of inoculations [16]. In this study, all animals were naïve, including their mothers. The goats had a high response to the first vaccination followed by a substantial decrease 1 year later, whereas for sheep, initial levels were lower with a slight elevation 1 year later only when the ewes received a booster. Regarding booster inoculation, in the current study, there was no significant effect on the antibody response 1 year later. One major outcome of this trial was the group size of tested animals, especially the remaining sheep and goats a year later. Since the study was performed on commercial farms, we were subjected to a dropout due to economic considerations. Still, experimental size was sufficient for a preliminary study. Concerning our preliminary examination of the passage of antibodies from ewes to their offsprings, we found that lambs rely on the absorption of antibodies (immunoglobulins, especially IgG) from the colostrum of their mothers right after birth. The efficiency of this transfer is crucial for lamb survival and immunity in the early stages of life [17]. The antibody level of the colostrum samples we examined was below the protective level, although a quarter of the kids had sufficient rabies antibody level in their serum, after receiving colostrum from a “bank colostrum.” Research shows that vaccination of ewes can influence the types and quantities of antibodies transferred to lambs. The timing and nature of vaccinations play a significant role in the amount of specific antibodies present in colostrum [18]. Thus, further study is warranted to gain insights in relation to the efficiency of antibody passage to the young, naïve animals.

## 5. Conclusions

The maintenance of an immunological response to rabies 1 year after an initial vaccination dose using a commercially available adjuvanted and killed-virus rabies vaccine for both sheep and goats was successful. In this study, and with this test group of animals, the booster made a differential contribution to the species’ antibody response: 1 year after the vaccine, 82% of the ewes that had received a booster dose remained protected, whereas 46% of the ewes that did not receive a booster were immunologically protected against rabies. Among the does, 83% of those receiving a booster maintained sufficient antibody levels 1 year later, whereas 80% of the does without a booster remained protected, demonstrating no significant additional contribution of the booster dose in this group of goats. Overall, the goats’ response appeared to be very different from that in sheep. The goats showed a much stronger initial response at 1 month and (despite falling titers) retained a high protection level at 1 year. In contrast, the sheep showed a lower response in terms of titers at 1 month (although protection was good, but significantly lower than in goats), and without a booster at 1 month, the protection level in sheep at 1 year was poor.

First, and most importantly, this small-scale study demonstrates the benefits of annual rabies vaccination for both species. It further demonstrates the benefit of a booster dose in sheep, but not in goats. In general, the study results are premature for evidence-based policy decisions that differ from annual vaccination recommendations; still, they reveal interesting differences in the immunological response between individuals and between species. In light of those initial findings, further, larger, and longer studies are warranted to recommend a robust vaccine protocol for each species.

## Figures and Tables

**Figure 1 vetsci-11-00502-f001:**
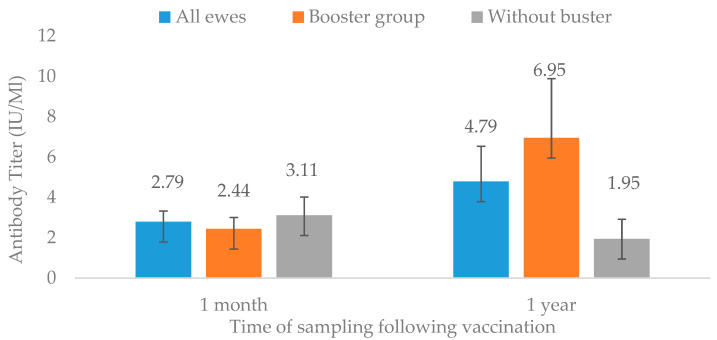
Antibody titer of sheep vaccinated with one dose and one dose + booster, 1 month and 1 year post rabies vaccination. Data are presented as mean ± SE. Values are in IU/mL. No statistical difference between results.

**Figure 2 vetsci-11-00502-f002:**
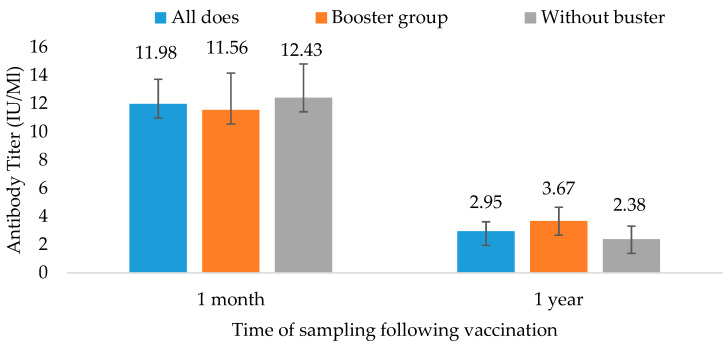
Antibody titer of goats vaccinated with one dose and one dose + booster, 1 month and 1 year post rabies vaccination. Data are presented as mean ± SE. Values are measured in IU/mL. No statistical difference between results.

**Figure 3 vetsci-11-00502-f003:**
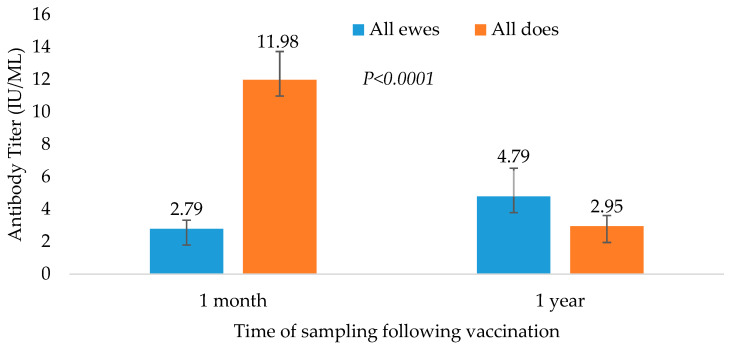
Comparison of the immunological responses of sheep and goats, 1 month and 1 year post rabies vaccination. The difference in immune response between species after one month was significant (*p =* 0.001). results for the following year are not significant.

**Table 1 vetsci-11-00502-t001:** Comparison of non-boosted sheep 1 month and 1 year post rabies vaccination.

Non-Boosted Sheep	1 Month	1 Year
Protected (n)	62	6
Not protected (n)	8	7
Protected * (%)	88.6%	46.2%

* Difference between protected sheep at 1 month and 1 year was significant (*p* = 0.002).

**Table 2 vetsci-11-00502-t002:** Comparison of non-boosted goats 1 month and 1 year post rabies vaccination.

Non-Boosted Goats	1 Month	1 Year
Protected (n)	49	12
Not protected (n)	0	3
Protected * (%)	100.0%	80.0%

* Difference between protected goats at 1 month and 1 year was significant (*p* = 0.02).

**Table 3 vetsci-11-00502-t003:** Cross-tab comparison of sheep and goats 1 month post rabies vaccination.

	Sheep	Goats
Protected (n)	62	49
Not protected (n)	8	0
Total (n)	70	49
Protected (%) *	88.6%	100.0%

* Difference between protected sheep and protected goats at 1 month was significant (*p* = 0.03).

**Table 4 vetsci-11-00502-t004:** Influence of animal species (sheep/goat), booster administration, and the interactions among the variables on the success of immune response to rabies vaccination 1 year post-vaccination.

Variables	Level ²	*n*	LSM Diff. ^1^	SE	*p*
Species	Ewes	30	Ref.	-	0.473
	Does	27	−1.84	2.2266	
Booster	0	28	Ref.	-	0.538
	1	29	3.42	2.1703	
Success	No	15	Ref.	-	0.037
	Yes	42	5.04	2.2658	
Species*Booster	Ewes*0	13	Ref.	-	1
	Ewes*1	17	−2.7208	2.796	
	Does*0	15	Ref.	-	1
	Does*1	12	0.03515	3.0904	
Species*Success	Ewes*No	10	Ref.	-	0.212
	Ewes*Yes	20	−6.1152	2.8256	
	Does*No	5	Ref.	-	1
	Does*Yes	22	−3.5621	3.5359	

^1^ LSM values are deviations from reference levels. Reference levels of species, booster, success, and interactions are 0.0, corresponding to antibody levels of 4.79, 2.18, 0.3 IU/mL, respectively. ² Levels: 0 represents animal which didn’t received booster, 1—represent animals which received booster. Yes/no—represent the success in protection levels. Reference levels of the variables correspond to probabilities of success of each variable.

## Data Availability

Some of the original data presented in the study are openly available on the website of Israel Ministry of Agriculture and Food Security (https://www.gov.il/en/departments/topics/kalevet/govil-landing-page, accessed on the 13 October 2024). Further information is available on request from the corresponding author.

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
