# Peer review of "Rabies Vaccination for Sheep and Goats: Influence of Booster on Persistence of Antibody Response"

_vetsci, 2024, doi:10.3390/vetsci11100502_

Round 1

Reviewer 1 Report

Comments and Suggestions for Authors

This paper briefly reported the benefits of annual rabies vaccination for both species in small-scale study. It further showed the immunoenhancement benefit of a booster dose in sheep, but not in goats. Moreover, this study demonstrated interesting differences in the immune response between individuals and between species, which implying that further, larger and longer experiments are needed for proper vaccination protocol for sheep and goats. Anyway, their results confirmed that the differences of immune system should be considered when rational rabies vaccination protocol are designed for different animals. This manuscript is generally acceptable for publication after minor revision.

1.     The authors should explain or add the exact adjuvant molecules in adjuvanted rabies vaccine.

2.     The sentence 218-219 seems to be wrong for the mean antibody level of 2.4±3.6 IU/ml in one dose vaccinated goats, which is not in accordance with Fig.2

3.     The authors failed to indicate the statistical analysis results among the three groups in the Fig.1, 2 or  between ewes and does in Fig.3, which should be added to clarify the immune response difference in different animals.  

Author Response

Reviewer no' 1:

The authors should explain or add the exact adjuvant molecules in adjuvanted rabies vaccine.

A: First, thank you for review.

The information about the adjuvant is written in the datasheet of the commercial vaccine.

"The adjuvants commonly known as “aluminium hydroxide are typically aluminium oxyhydroxide salts, which are usually at least partially crystalline. Aluminium oxyhydroxide, which can be represented by the formula AlO (OH), can be distinguished from other aluminium com pounds, such as aluminium hydroxide Al(OH), by infrared (IR) spectroscopy, in particular by the presence of an adsorp tion band at 1070 cm and a strong shoulder at 3090-3100 cm."

Can be retrieved from the website: Qualitative and quantitative composition - Rabisin (noahcompendium.co.uk).

  1. The sentence 218-219 seems to be wrong for the mean antibody level of 2.4±3.6 IU/ml in one dose vaccinated goats, which is not in accordance with Fig.2

A: I agree, numbers after the dot should be fully entered and uniformity is important. It was corrected.

  1. The authors failed to indicate the statistical analysis results among the three groups in the Fig.1, 2 or  between ewes and does in Fig.3, which should be added to clarify the immune response difference in different animals.  

A: it was corrected.

Reviewer 2 Report

Comments and Suggestions for Authors

The study analyses the effect of pre-exposure vaccination and booster on the persistence of antibody response in goat and sheep populations. The authors selected ewe and does for preselected farms and measured the antibody response to the first dose of vaccine and booster dose 30 days after the initial vaccination. Although the manuscript tries to answer an important public health and veterinary concern related to rabies exposure, there are several concerns related to the manuscript that needs to be addressed. My main concern is with the sample size, especially in estimating the effect size of the immune response one year after the booster dose administration, as there was a significant reduction in sample size in the follow-up period.

Major comments:

1.      Abstract:

o   The introduction and background of the abstract is vague. Suggestion: Briefly provide the background information on how exposure of livestock especially sheep and goats to rabid wildlife/dogs is a concern in Israel, which is the main focus of the paper.

o   Methodology: What kind of statistical analysis was performed and what was measured using those tests?

o   Provide quantitative results in the form of effect size, uncertainties and p-values and not just as percentages

2.      Line 145: Why were the two serological tests conducted ie., rabies fluorescent focus inhibition test and rapid immunofluorescent antibody test? What are the differences between them? Which test result was used in the final analysis?

3.      Please explain want “proc glimmix” procedure is for the readers who are not familiar with SAS-specific products. Why was it chosen for this specific analysis?

4.        Fisher's exact test is a statistical test for categorical variables. What were the categorical variables here (ie.  the outcome and input variables)?

5.      Line 162: Please specify what figures and tables are you referring to. It seems like you are referring to the ones presented in the results. If so, please move these sentences to the result section instead.

6.      Methodology: how was the sample size calculated? Was the sample size big enough to estimate effect size both during the first and second titer checks?

7.      Figures 1 and 2:

o   The Lower limit for the error bars is not visible for all groups. Please correct this.

o   The legend description: “Without booster” instead of “Without buster”

8.      Table 1 and 2: Only 13 out of 70 ewes (and 15 out of 49 does) that were included in the study initially were followed up to test the effect of booster dose. Though a significant difference was observed, was the sample size big enough to test this effect size? Was power analysis conducted?

9.       Discussion: The discussion needs major revision. Discussion should not be the reiteration of the result section. Rather, it's essential to interpret your findings, compare them to previous research, and address their broader implications. Here are some suggestions on how to restructure it.

o   Summarize the key findings.

o   Conduct a literature review and compare your findings with the previously published research

o   Talk about the limitations and strengths of your fin

o   Address the practical implications of your findings for public health practice and policy

o   Include more references.

10.  The authors state that RFFIT and the specificity of only 89%. Could this have affected the overall results? Please discuss.

11.   Though stated as the main contribution, it is not clear how the study contributed to “better understand the immunological response to the vaccine and recommend a more adequate vaccination protocol” as stated in Line 99.

Minor comments:

1.      Figure 1, 2, 3:  Representing the figures in the form of box and whisker plots with each data point visible would be a better of representing that bar plot.

2.      Line 96: Please be specific. What does “different rabies vaccination protocol mean”

3.      Line 56: Please provide more details on what “carnivores” means. Does it mean wild-life species? How does an outbreak in domestic species occur? Is it due to the exposure of livestock to wildlife?

4.      Line 144: Please specify what samples were tested using rabies fluorescent focus inhibition test

Comments on the Quality of English Language

No comments

Author Response

ראש הטופס

Major comments:

  1. Abstract:

o   The introduction and background of the abstract is vague. Suggestion: Briefly provide the background information on how exposure of livestock especially sheep and goats to rabid wildlife/dogs is a concern in Israel, which is the main focus of the paper.

A: First, I would like to thank you for your thorough review.

As I read the beginning of the abstract, I understood your remark and accepted it. It was corrected.

o   Methodology: What kind of statistical analysis was performed and what was measured using those tests?

A: It is elaborated in the "statistical analysis" section: "Differences in protection rates by different factors were tested using the Fisher exact test (two-tailed). The factors associated with antibody level after 1 month and 1 year were analyzed in multifactorial models using the proc glimmix procedure of SAS (version 9.2, SAS Institute). The statistical model included antibody level after 1 month or after 1 year (success) = species + booster + species*booster; where species = ewes (sheep) or does (goats), booster = receive or did not receive, success = level of antibody higher or lower than 0.5 IU/ml." To compare levels within a variable, we ran the Bonferroni adjustment for multiple comparisons

o   Provide quantitative results in the form of effect size, uncertainties and p-values and not just as percentages.

  1. Line 145: Why were the two serological tests conducted ie., rabies fluorescent focus inhibition test and rapid immunofluorescent antibody test? What are the differences between them? Which test result was used in the final analysis?

A: It's the same test. This sentence was corrected. We meant the RFFIT (Rapid Fluorescent Focus Inhibition Test).

  1. Please explain want “proc glimmix” procedure is for the readers who are not familiar with SAS-specific products. Why was it chosen for this specific analysis?

A: PROC GLIMMIX is used for for Binary Variables When dealing with binary variables (variables with only two possible values, like yes/no or 0/1).

In this study we test the probability of the success of the vaccine after 1 month and after 1 year according to the antibody level using a specific cutoff level for success or unsuccess. We chose this procedure because we have binary variables (yes or no).

  1. Fisher's exact test is a statistical test for categorical variables. What were the categorical variables here (ie.  the outcome and input variables)?

A: The Fisher's exact test was used to measure the difference between protected and not protected animals. We test the difference after 1 month and after 1 year.  The categorical variables were success or not success according to level of Antibodies.

  1. Line 162: Please specify what figures and tables are you referring to. It seems like you are referring to the ones presented in the results. If so, please move these sentences to the result section instead.

A: In the Statistical Analysis section, we presented the model and what we tested. The figures are in the result section and there you can find also the description of the result.

  1. Methodology: how was the sample size calculated? Was the sample size big enough to estimate effect size both during the first and second titer checks? Small data and limitation

A: The study is preliminary study to test the effect of the vaccine. We did it in two different farms. So, there is limited amount of animals and has you can see we didn’t have many animals especially after 1 year. We are familiar with the problem that this is small-scale study and we need to have larger studies. In the discussion part we talk about this limitation. But with this small-scale study there are interesting result that can benefit for the scientific community.   

  1. Figures 1 and 2:

o   The Lower limit for the error bars is not visible for all groups. Please correct this.

A: It was corrected.

o   The legend description: “Without booster” instead of “Without buster”

A: We intended to keep the "booster" version throughout all the text and legends.

  1. Table 1 and 2: Only 13 out of 70 ewes (and 15 out of 49 does) that were included in the study initially were followed up to test the effect of booster dose. Though a significant difference was observed, was the sample size big enough to test this effect size? Was power analysis conducted?

A: Since we used for the study two commercial farms, a part of our experimental animals were excluded due to commercial reasons, and we had no ability to foresee it or prevent it. It is not optimal but group size was within statistical limits.  

  1. Discussion: The discussion needs major revision. Discussion should not be the reiteration of the result section. Rather, it's essential to interpret your findings, compare them to previous research, and address their broader implications. Here are some suggestions on how to restructure it.

o   Summarize the key findings.

o   Conduct a literature review and compare your findings with the previously published research

o   Talk about the limitations and strengths of your fin

o   Address the practical implications of your findings for public health practice and policy

A: I agree. I have summarized the key findings, and I have inserted previous finding in other trials and I have enlarged the discussion including comparing of the results, mentioning the strength and limitation of this study, and overall your suggestions were added.

o   Include more references.

A: added.

  1. The authors state that RFFIT and the specificity of only 89%. Could this have affected the overall results? Please discuss.

A: The RFFIT (Rapid Fluorescent Focus Inhibition Test) is a widely used method for assessing the effectiveness of rabies vaccination in humans and animals. It is designed to measure the presence of rabies virus-neutralizing antibodies in the blood, which are an indicator of how well a vaccine is working. The specificity of the RFFIT is generally very high, often approaching 100%, meaning it is very effective at correctly identifying non-positive (negative) results for rabies antibodies. Sensitivity is also high, usually in the range of 95-100%, depending on the quality of the test and the standards used. This means it can accurately detect positive results (presence of rabies-neutralizing antibodies) in a large proportion of cases where they are actually present. Overall, this method is considered gold standard, widely published in WHO and WOAH (formal OIE). All samples were tested with the same method, maintaining uniformity. The RFFIT is a valid and reliable method, used worldwide.

  1. Though stated as the main contribution, it is not clear how the study contributed to “better understand the immunological response to the vaccine and recommend a more adequate vaccination protocol” as stated in Line 99.

A: The sentence was corrected. We meant that the immunological response of the goats and sheep is better understood due to this preliminary work.

Minor comments:

  1. Figure 1, 2, 3:  Representing the figures in the form of box and whisker plots with each data point visible would be a better of representing than bar plot.

A: We think it’s a question of personal style, or preferences. We are used to work with graphs in the form of box, we think its better understandable. We would like to leave it like that, with your permission.

  1. Line 96: Please be specific. What does “different rabies vaccination protocol mean”

A: Corrected.

  1. Line 56: Please provide more details on what “carnivores” means. Does it mean wild-life species? How does an outbreak in domestic species occur? Is it due to the exposure of livestock to wildlife?

 A: We meant to sylvatic rabies, it was corrected. As we wrote later on, in that chapter, stray dogs are also involved, probably by exposure to wildlife carnivores.

  1. Line 144: Please specify what samples were tested using rabies fluorescent focus inhibition test

A: all samples were tested using the Rabies Fluorescent Focus Inhibition Test (RFFIT).

Comments on the Quality of English Language

No comments

Submission Date

04 August 2024

Date of this review

21 Aug 2024 03:10:56

תחתית הטופס

© 1996-2024 MDPI (Basel, Switzerland) unless otherwise stated

Reviewer 3 Report

Comments and Suggestions for Authors

The authors have evaluated the response of rabies vaccine in sheep and goat by evaluating the induction of neutralizing antibodies.

The authors have used well established  experimental design for determining the efficacy of rabies vaccine in animals. The conclusions drawn from the analysis appear reasonable.

Minor comments

a) Authors have used the term like protected and unprotected inaugurating throughout the manuscript like----- "protected groups"---line 182; "unprotected ewes"------190. The titers may be enough to provide protection, but authors have not proved it.

b )Appears missplelled;   "does"--------line 241;270 etc

Author Response

Minor comments

  1. Authors have used the term like protected and unprotected inaugurating throughout the manuscript like----- "protected groups"---line 182; "unprotected ewes"------190. The titers may be enough to provide protection, but authors have not proved it.

A: First, thank you for your review. The term "protective levels" is commonly used in areas of vaccine studies terminology. However, "protected groups" may be less suitable, so I agree, and corrected it.

b )Appears missplelled;   "does"--------line 241;270 etc

A: meaning- "does" the female goats.